# Screening the pandemic response box identified benzimidazole carbamates, Olorofim and ravuconazole as promising drug candidates for the treatment of eumycetoma

**Wilson Lim**[1], **Bertrand Nyuykonge**[1], **Kimberly Eadie**[1], **Mickey Konings**[1], **Juli Smeets**[1], **Ahmed Fahal**[2], **Alexandro Bonifaz**[3], **Matthew Todd**[4], **Benjamin Perry**[5], **Kirandeep Samby**[6], **Jeremy Burrows**[6], **Annelies Verbon**[1], **Wendy van de Sande**[1] *

**1** Erasmus MC, University Medical Center Rotterdam, Department of Microbiology and Infectious Diseases, Rotterdam, The Netherlands, **2** Mycetoma Research Centre, University of Khartoum, Khartoum, Sudan, **3** Hospital General de Mexico Dr Eduardo Liceaga, Mexico City, Mexico, **4** University College London, School of Pharmacy, London, United Kingdom, **5** Drugs for Neglected Diseases initiative (DNDi), Geneva, Switzerland, **6** Medicines for Malaria Venture (MMV), Geneva, Switzerland

* w.vandesande@erasmusmc.nl

**Data Availability Statement:** All relevant data are within the manuscript and its Supporting

## Abstract

Eumycetoma is a chronic subcutaneous neglected tropical disease that can be caused by more than 40 different fungal causative agents. The most common causative agents produce black grains and belong to the fungal orders Sordariales and Pleosporales. The current antifungal agents used to treat eumycetoma are itraconazole or terbinafine, however, their cure rates are low. To find novel drugs for eumycetoma, we screened 400 diverse drug-like molecules from the Pandemic Response Box against common eumycetoma causative agents as part of the Open Source Mycetoma initiative (MycetOS). 26 compounds were able to inhibit the growth of *Madurella mycetomatis*, *Madurella pseudomycetomatis* and *Madurella tropicana*, 26 compounds inhibited *Falciformispora senegalensis* and seven inhibited growth of *Medicopsis romeroi in vitro*. Four compounds were able to inhibit the growth of all five species of fungi tested. They are the benzimidazole carbamates fenbendazole and carbendazim, the 8-aminoquinolone derivative tafenoquine and MMV1578570. Minimal inhibitory concentrations were then determined for the compounds active against *M. mycetomatis*. Compounds showing potent activity *in vitro* were further tested *in vivo*. Fenbendazole, MMV1782387, ravuconazole and olorofim were able to significantly prolong *Galleria mellonella* larvae survival and are promising candidates to explore in mycetoma treatment and to also serve as scaffolds for medicinal chemistry optimisation in the search for novel antifungals to treat eumycetoma.

## Author summary

Mycetoma is a neglected tropical disease characterised by the formation of tumorous swellings and the presence of grains. In fungal mycetoma (eumycetoma), the most

Information files. Additional results can be found on https://github.com/OpenSourceMycetoma.

**Funding:** The work performed on the Pandemic Response Box by MMV presented in this manuscript is supported by funding from the The Bill & Melinda Gates Foundation's Open Access Policy. Full details of MMV funding can be found https://www.mmv.org/about-us/funding-and-expenditure. The funders had no role in study design, data collection and analysis, decision to publish, or preparation of the manuscript.

**Competing interests:** The authors have declared that no competing interests exist.

common causative agents produce black grains although genetically, these fungi can be very different. *Madurella mycetomatis*, *Madurella pseudomycetomatis* and *Madurella tropicana* belong to the fungal order Sordariales, while *Falciformispora senegalensis* and *Medicopsis romeroi* belong to the order Pleosporales. Treatment for eumycetoma is challenging and antifungal therapy with itraconazole or terbinafine is combined with surgery. Unfortunately, cure rates of only 26% are obtained and amputation of the affected area is often needed. Despite the urgent need to find new antifungals for the treatment of eumycetoma, only fosravuconazole is in the pipeline to treat mycetoma. To discover novel compounds with activity against eumycetoma causative agents, the Open Source Mycetoma (MycetOS) initiative was founded. As part of this initiative, we previously tested 800 compounds from the Pathogen Box and Stasis box for their efficacy against *M. mycetomatis*. In this study, we have tested 400 compounds from the Pandemic Response Box against *Madurella mycetomatis*, *Madurella pseudomycetomatis*, *Madurella tropicana*, *Falciformispora senegalensis* and *Medicopsis romeroi*. We have identified four compounds that were able to inhibit all five fungi species *in vitro*, namely fenbendazole, carbendazim, tafenoquine and MMV1578570. Fenbendazole, MMV1782387, ravuconazole and olorofim were also able to significantly prolong larvae survival in our *in vivo Galleria mellonella* model. This study showed benzimidazole carbamates as promising candidates to further explore for eumycetoma treatment.

## Introduction

Mycetoma is a chronic subcutaneous neglected tropical disease commonly found in tropical and sub-tropical regions [1]. It commonly affects the lower extremities and is characterised by tumorous swellings and the excretion of pus and grains [1]. These grains are small aggregates of the causative agent that are embedded in a protective cement material. Mycetoma can be caused by more than 70 different causative agents and is categorized into actinomycetoma (caused by bacteria) and eumycetoma (caused by fungi). Eumycetoma can be caused by more than 40 different fungal causative agents which produce either black or white grains [2]. The most common causative agents in eumycetoma produce black grains and belong to the fungal orders Sordariales and Pleosporales. *Madurella mycetomatis* of the order Sordariales is the most common eumycetoma causing agent representing 75.1% of cases worldwide, next in line is *Falciformispora senegalensis* from the order Pleosporales at 6.2% of cases worldwide [3]. The prevalence of these species differs per country.

Treatment options for mycetoma are dependent on the causative agent. In general, actinomycetoma is treated with antimicrobials with a high success rate [1]. For eumycetoma, a combination of surgery and prolonged medication is necessary [4,5]. Amputation of the affected parts is common when treatment fails. The current antifungal agent used to treat eumycetoma is itraconazole at 400 mg/day for six months, followed by surgery and then another 400 mg/day for at least six more months[1]. Clinical responses to itraconazole are often variable and are associated with recurrences even after extended treatment periods and surgery [1,6]. The cure rate of itraconazole can differ between studies and is generally low with only 8% - 26% of patients cured [7,8]. Next to itraconazole, some countries also use the antifungal agent terbinafine in combination with surgery. In a report from Senegal, patients were given terbinafine 500 mg twice daily for 24–48 weeks combined with surgery. A cure rate of 30% similar to that of itraconazole was noted [9]. Half of the patients that were cured underwent surgical removal of the affected area, while the other half had amputations of the affected parts [9]. The

disappointing cure rates of itraconazole and terbinafine exhibits the urgent need to find a new drug for eumycetoma treatment.

To discover new drug candidates to treat mycetoma, we have previously tested 800 compounds from the Pathogen Box and Stasis Box obtained from Medicines for Malaria Venture (MMV) for activity against *M. mycetomatis* [10]. These compound libraries contained drug-like molecules previously shown to be active against pathogens causing tropical and neglected diseases (Pathogen Box) or candidates that had been studied in clinical studies and could, thus be potentially repurposed for neglected diseases (Stasis Box). These were made available by MMV as an open access initiative tool to stimulate research and development in neglected diseases [11,12]. In return, researchers were asked to share their findings in the public domain, creating an open and collaborative forum for infectious disease drug research. Out of 800 compounds screened, we discovered 215 compounds that were able to inhibit *M. mycetomatis* growth at a concentration of 100 μM *in vitro* and five that were able to prolong larvae survival in an *in vivo Galleria mellonella* wax moth model [10]. That resulted in the discovery of fenarimols as a potential new class of antifungal compounds able to inhibit *M. mycetomatis* growth both *in vitro* and *in vivo* [10]. Eumycetoma however, can also be caused by fungi other than *M. mycetomatis*, therefore, it is crucial to know if these compounds can also inhibit other common causative agents of mycetoma.

In 2019, MMV and Drugs for Neglected Diseases Initiative (DND*i*) launched a new Open Access compound box called the Pandemic Response Box. This box contains 400 diverse drug-like molecules active against bacteria, viruses and fungi. Similar to the Pathogen and Stasis box, it was also created to stimulate research and development in neglected diseases. To discover new compounds able to inhibit eumycetoma causative agents, we decided to screen this box against *M. mycetomatis* and other common black-grain eumycetoma causative agents. For this we selected the *Madurella* sibling species (*M. mycetomatis*, *Madurella pseudomycetomatis* and *Madurella tropicana*) from the order of the Sordariales and *F. senegalensis* and *Medicopsis romeroi* from the order of the Pleosporales (Fig 1). To identify compounds able to inhibit the growth of these five eumycetoma causative agents, the Pandemic Response Box was first screened *in vitro*. Compounds with the promising activity were further tested *in vivo* in our *M. mycetomatis* grain model in *Galleria mellonella* larvae to determine their efficacy.

## Materials and methods

### Chemical libraries

The Pandemic Response Box was kindly provided by Medicines for Malaria Venture (MMV, Geneva, Switzerland). Within this box, compounds were present in a concentration of 2 mM or 10 mM in DMSO. The list of compounds included in this box can be found in the S1 Table and at the Pandemic Response Box website (https://www.mmv.org/mmv-open/pandemic-response-box).

### Fungal isolates

*M. mycetomatis* genome strain MM55, *M. pseudomycetomatis* strain Mex2A; *M. tropicana* strain CBS206.47; *F. senegalensis* strain CBS132257 and *Me. romeroi* strain CBS128765 were used to identify compounds able to inhibit the growth at 25 μM. *M. mycetomatis* strain MM55 was used to determine the IC50 of these compounds. To determine the concentration that inhibited 50% of *M. mycetomatis* isolates (MIC50), the minimal inhibitory concentration of these compounds was determined in ten different *M. mycetomatis* isolates namely, AL1, CBS 247.48, I1, I11, MM14, MM45, MM55, P1, Peru72012 and SO1. The fungal isolates were obtained from the Mycetoma Research Center in Sudan, Hospital General de Mexico

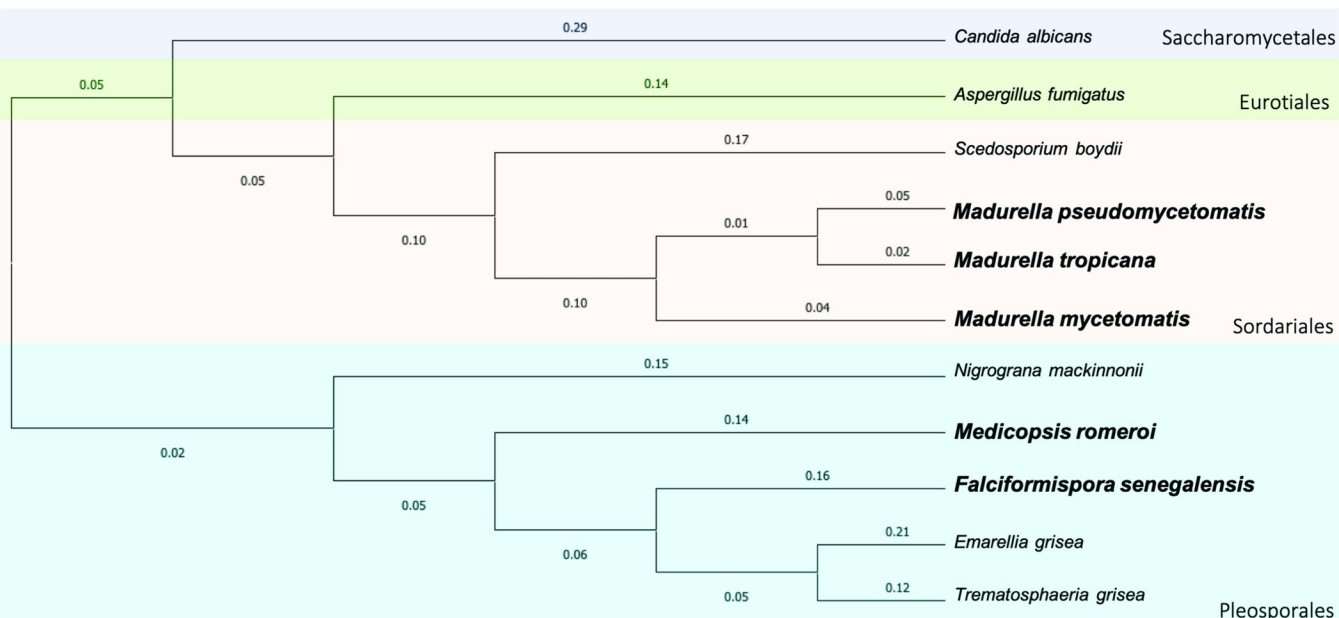

**Fig 1. Phylogenetic tree of the most common eumycetoma causing agents.** *Madurella pseudomycetomatis*, *Madurella mycetomatis* and *Madurella tropicana* belong to the order Sordariales, while *Medicopsis romeroi* and *Falciformispora senegalensis* belong to the order Pleosporales. Bolded characters indicate the fungal species used in this evaluation.

Dr Edurado Liceaga in Mexico, and Westerdijk Fungal Biodiversity Institute in the Netherlands. Isolates are maintained in Erasmus Medical Centre. All isolates were identified to the species level based on morphology and sequencing of the ITS regions [13]. The *M. mycetomatis* were genetically diverse, and were shown to have unique *Mmy*STR genotypes [14].

## Phylogenetic analysis of fungal isolates

Phylogenetic tree analysis was performed using Molecular Evolutionary Genetics Analysis (Mega X) (Pennsylvania State University, USA) on isolates *M. mycetomatis* (DQ836767.1), *M. pseudomycetomatis* (MN545597.1), *M. tropicana* (JX280869.1), *Me. romeroi* (MH865072.1), *F. senegalensis* (MH861197.1), *Trematosphaeria grisea* (NR_132039.1), *Emarellia grisea* (LT726708.1), *Nigrogana mackinnonii* (MG063816.1), *Scedosporium boydii* (MH864818.1), *Aspergillus fumigatus* (NR_121481.1) and *Candida albicans* (NR_125332.1) retrieved from GenBank. Alignment was performed using ClustalW and a phylogenetic tree was constructed using a maximum likelihood estimation.

## Screening the pandemic response box

To determine which of the compounds present in the Pandemic Response Box were able to inhibit *M. mycetomatis*, *M. pseudomycetomatis*, *M. tropicana*, *F. senegalensis* and *Me. romeroi* growth, the CLSI M38-A2 based *in vitro* susceptibility assay for eumycetoma causative agents was used. In the eumycetoma causative agents optimized protocol, fungi were cultured for ten days at 37˚C in RPMI 1640 medium supplemented with L-glutamine (0.3 g/litre) and 20 mM morpholinepropanesulfonic acid (MOPS). The mycelia were harvested by a 5-min centrifugation and were washed with sterile saline. To homogenize the inoculum, the mycelia were sonicated for 20 s at 28 μm (Soniprep, Beun de Ronde, The Netherlands). The final inoculum was

prepared from the homogenized fungal suspension mixed with RPMI medium to obtain a transmission of 70% at 660 nm (Novaspec II; Pharmacia Biotech) [15,16]. The screening procedure was performed in 96 well microplates and 2,3-bis(2-methoxy-4-nitro-5-sulfophenyl)-5-[(phenylamino)carbonyl]-2H-tetrazolium hydroxide (XTT) or resazurin was used to facilitate end-point reading [10,17]. Initial screening of the compounds was performed at the concentration of 25 µM. Compounds that inhibited >75% (resazurin) or >80% (XTT) fungal growth were selected. To determine the concentration at which 50% reduction in fungal growth (IC50) was observed, a hyphal suspension of *M. mycetomatis* MM55 was incubated with compounds at a 2-fold dilution series ranging from 16 µM to 0.03125 µM. The IC50 was then determined by plotting the growth percentage at fixed concentrations and determining the concentration at which 50% reduction of growth was obtained. To determine the minimal inhibitory concentration of the compounds, the minimal inhibitory concentrations (MIC) were calculated. MIC was defined as the concentration at which 80% or more reduction in metabolic activity was obtained as determined by XTT [15]. Metabolic activity was calculated using the formula $(E_{sample}-E_{nc})/(E_{gc}-E_{nc})^*100\%$ and measured colormetrically at 450 nm. For resazurin, percentage growth was calculated using the formula $(E_{nc}-E_{sample})/(E_{nc}-E_{gc})^*100\%$ where nc is the negative control and gc is the growth control. Resazurin was measured colormetrically at 600 nm. To determine the MIC, a 2-fold dilution series ranging from 16 µM to 0.03125 µM was prepared for each compound. The median MIC of a compound over ten *M. mycetomatis* isolates is referred to as MIC50.

## Toxicity and infection in *G. mellonella* larvae grain model

To determine the toxicity of the identified compounds in *G. mellonella* larvae, a single dose of 20 µM per compound was injected in the last pro-leg. Survival was monitored for ten days. A compound was considered non-toxic if no significant difference between the control and treated larvae was determined. Compounds that were not toxic for *G. mellonella* larvae were further used in infection studies to determine their activity against *M. mycetomatis* according to our previously published protocol [10,18]. In short, *M. mycetomatis* isolate MM55 mycelia were cultured in colourless RPMI 1640 medium supplemented with L-glutamine (0.3 g/L), 20 mM mopholinepropanesulfonic acid (MOPS) and chloramphenicol (100 mg/L; Oxoid, Basingstoke, United Kingdom) for 2 weeks at 37˚C and sonicated for 2 minutes at 28 microns. The resulting homogenous suspension was washed once in PBS and further diluted to an inoculum size of 4 mg wet weight per larvae. Inoculation was performed by injecting 40 µL of the fungal suspension in the last left pro-leg with an insulin 29 G U-100 needle (BD diagnostics, Sparsk, Nevada, USA). Larvae were treated with 20 µM of compound per larvae and controls were injected with solvent. Compounds were administered 4, 28, and 52 hours after infection. Treatment was started at four hours post-infection since at that time point grains were already present in larvae. In this infection model, larvae are treated only during the first three days after infection, after which a seven-day observation period without antifungal treatment starts. Larvae were monitored over ten days with their survival recorded on day four and day ten. If during these ten days a larvae would form a cocoon, they were left out of the equation, since it is not ascertainable if these individual larvae would have survived or perished during infection.

## Statistical analysis

To compare survival curves, the Log-rank test was performed with GraphPad Prism 7 (GraphPad Inc.) A p-value smaller than 0.05 was deemed significant.

## Results

### Screening the pandemic response box against five eumycetoma causative agents demonstrated that MMV003143 (fenbendazole), MMV1578570, MMV344625 (carbendazim) and MMV000043 (tafenoquine) were able to inhibit the growth of all five species at a concentration of 25 μM

In total, 400 diverse-drug like compounds from the Pandemic Response box were tested *in vitro* for their potency against *M. mycetomatis*, *M. pseudomycetomatis*, *M. tropicana*, *F. senegalensis* and *M. romeroi*. All associated data can be found on our online database on GitHub (https://github.com/OpenSourceMycetoma). Out of 400 compounds screened at the concentration of 25 μM, 47, 42 and 37 compounds were able to inhibit the growth of the Sordariomycetes *M. mycetomatis*, *M. pseudomycetomatis* and *M. tropicana*, respectively. 26 compounds were able to inhibit the growth of *Madurella mycetomatis*, *Madurella pseudomycetomatis* and *Madurella tropicana* together. 26 compounds inhibited *F. senegalensis* growth while only seven compounds inhibited growth of *Me. romeroi* (Figs 2 and 3). The compounds that inhibited growth of these five fungal species are spread among the antifungal, antibacterial and antiviral compound sets in the Pandemic Response Box, with most compounds fall under the antifungals and antibacterials, less than 21% were antivirals (Fig 3A). In total, only four compounds were able to inhibit the growth of all five fungal species tested (Fig 3B); they are MMV003143 (fenbendazole), MMV1578570, MMV344625 (carbendazim) and MMV000043 (tafenoquine). Strikingly, itraconazole and the other azoles were only found to inhibit the growth of the *Madurella* species, not the Pleosporalean fungi. Terbinafine on the other hand was only able to inhibit the growth of the Pleosporalean fungi, not the *Madurella* species (Table 1).

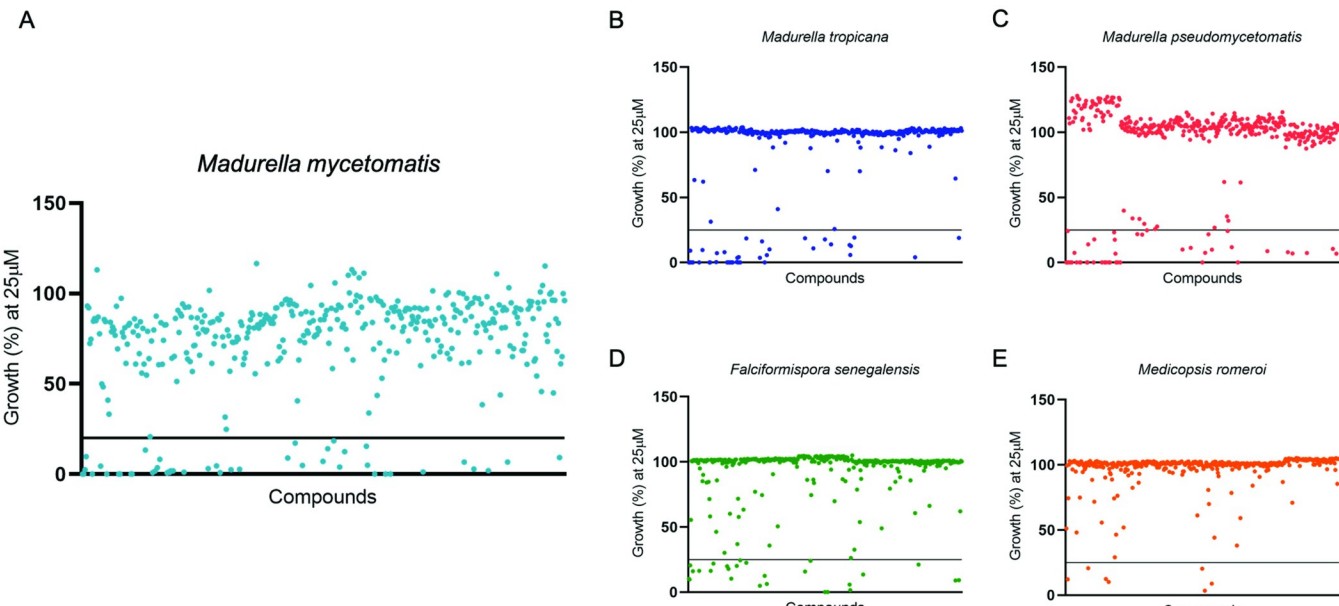

**Fig 2. Compounds inhibiting growth of the Sordariomycetes and the Pleosporaleans at 25 μM.** Panel A, *Madurella mycetomatis*; B, *Madurella tropicana*; C, *Madurella tropicana*; D, *Falciformispora senegalensis*; and E, *Medicopsis romeroi*. 47, 42, 37, 26 and 7 compounds were able to inhibit the tested fungal isolates respectively. The horizontal black-lines in the figures shows the growth percentage at 20%. Compounds situated under the black-lines were able that inhibited more than 80% fungal growth and thus a growth below 20%.

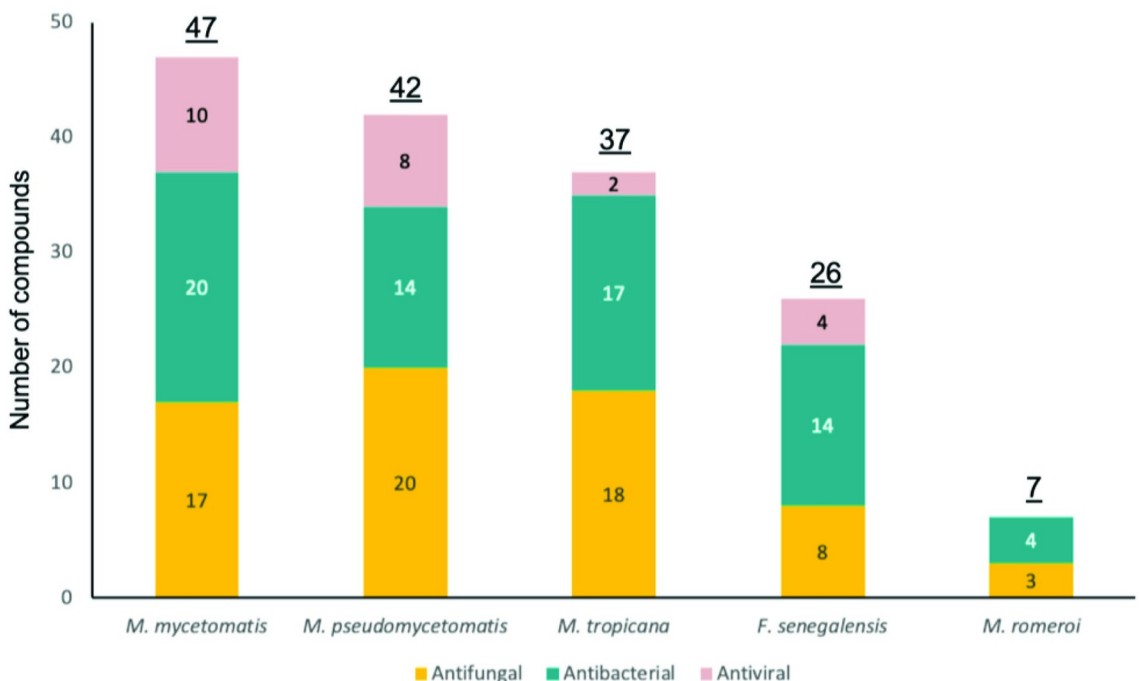

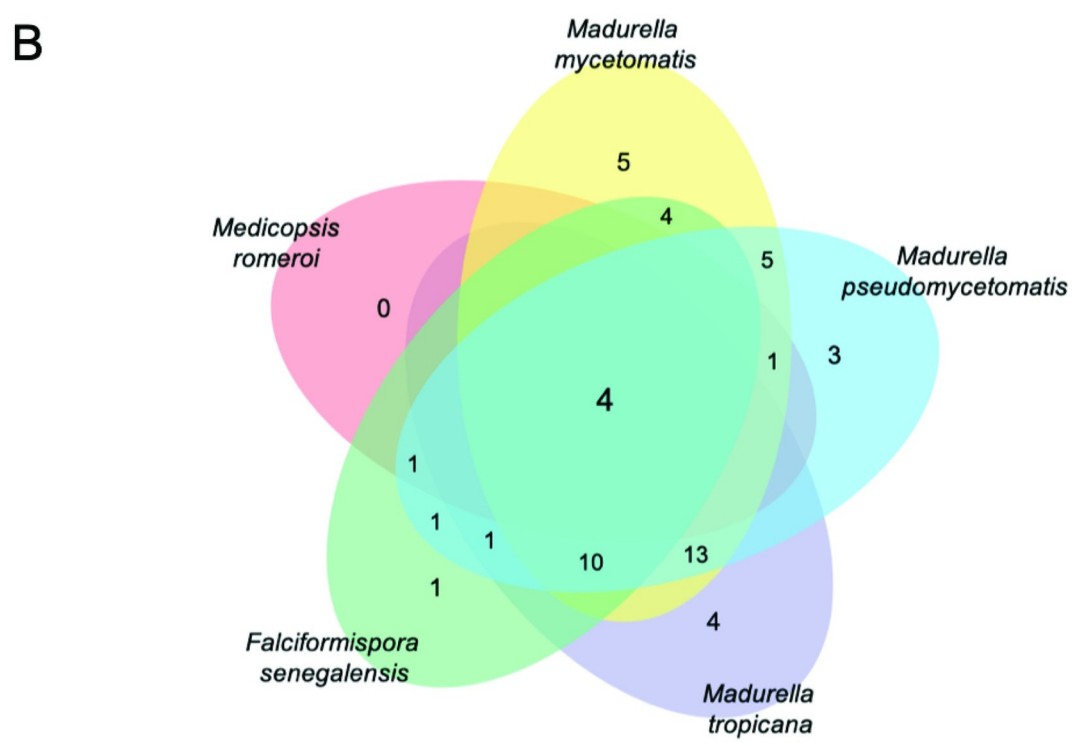

**Fig 3. Pandemic response box compounds exhibiting potency at 25 μM against the tested fungal species.** Panel A divides the compounds into their respective groups as indicated in the Pandemic Response Box. Most compounds exhibiting potency at 25 μM falls under the antifungals and antibacterial. Less than 21% are antivirals. No antivirals inhibit *M. romeroi* growth at 25 μM. Panel B displays a Venn diagram. Four compounds–Fenbendazole, Carbendazim, Tafenoquine and MMV1578570 were able to inhibit all 5 fungal species tested here.

## *In vitro* screening of the pandemic response box against *M. mycetomatis*

Since *M. mycetomatis* is substantially the most common causative agent of eumycetoma, all subsequent evaluations focused on this causative agent. To identify the most potent of the 45 compounds able to inhibit *M. mycetomatis* growth, their IC50 and IC90 values were determined. 20 compounds had an IC50 value below 13 μM (Table 2) with a median of at 2.66 μM (<0.1–12.8). Compounds with IC50 values higher than 13 μM were left out from further evaluation. In total, nine out of these 20 compounds targeted the ergosterol biosynthesis pathway. Among the nine compounds that targeted the ergosterol biosynthesis pathway, eight of them

**Table 1. Compounds from the Pandemic Response Box that inhibits the Sordariomycetes and the Pleosporaleans under 25 μM.**

| Madurella mycetomatis Madurella pseudomycetomatis Madurella tropicana | Falsiformispora senegalensis | Medicopsis romeroi |
|---|---|---|
| Carbendazim * | Carbendazim * | Carbendazim * |
| Fenbendazole * | Fenbendazole * | Fenbendazole * |
| Tafenoquine * | Tafenoquine * | Tafenoquine * |
| MMV1578570 * | MMV1578570 * | MMV1578570 * |
| Ciclopirox ¤ | Ciclopirox ¤ | Terbinafine ▲ |
| Eberconazole ¤ | Eberconazole ¤ | MMV1634402 ▲ |
| Ketoconazole ¤ | Ketoconazole ¤ | MMV1634399 |
| Miconazole ¤ | Miconazole ¤ | |
| MMV019724 ¤ | MMV019724 ¤ | |
| MMV1581548 ¤ | MMV1581548 ¤ | |
| MMV1593539 ¤ | MMV1593539 ¤ | |
| MMV1633966 ¤ | MMV1633966 ¤ | |
| MMV1634491 ¤ | MMV1634491 ¤ | |
| MMV1782140 ¤ | MMV1782140 ¤ | |
| MMV1782387 ¤ | MMV1782387 ¤ | |
| Abafungin | Terbinafine ▲ | |
| Alexidine | MMV1634402 ▲ | |
| Amorolfine | DNDI1417411 | |
| Isavuconazonium | Fludarabine | |
| Itraconazole | NSC 84094 | |
| Luliconazole | Ozanimod | |
| Olorofim | MMV1581545 | |
| OSU-03012 | MMV1582496 | |
| Ravuconazole | MMV1582497 | |
| SMR000040087 | MMV1593535 | |
| MMV1634386 | MMV1633963 | |

* Compounds inhibiting all 5 fungi.

¤ Compounds inhibiting *Madurella mycetomatis*, *Madurella pseudomycetomatis*, *Madurella tropicana* and *Falciformispora senegalensis*.

▲ Compounds inhibiting *Falciformispora senegalensis* and *Medicopsis romeroi*.

**Table 2. Class, mode of action, IC50 and MIC50 values of the 20 most potent compounds in the Pandemic Response Box against *Madurella mycetomatis*.** Compounds with MIC50 value indicated with an asterisk (\*) were determined in previous works [19–21]. Their efficacy to *M. mycetomatis* is again demonstrated here with their low IC50 values.

| Compounds | Trivial name or CHEMBL code | Class | Mode of Action | Use | IC50 (μM) | MIC50 (range) (μM)[1] |
|---|---|---|---|---|---|---|
| MMV1634362 | Ravuconazole | Azole | CYP51 inhibitor | Antifungals (humans)[N] | 0.01 | 0.01 (0.004–0.06) \* |
| MMV1782354 | Olorofim | Orotomides | DHODH inhibitor | Antifungals (humans)[O, S] | < 0.016 | 0.06 (0.004–0.13) \* |
| MMV1782224 | Luliconazole | Azole | CYP51 inhibitor | Antifungals (humans)[T] | < 0.03 | <0.03 (≦0.03) |
| MMV1634494 | Isavuconazonium | Azole | CYP51 inhibitor | Antifungals (humans)[O, S] | 0.04 | 0.04 (<0.02–0.17) \* |
| MMV637533 | Ketoconazole | Azole | CYP51 inhibitor | Antifungals (humans)[N] | 0.07 | 0.13 (0.06–1.9) \* |
| MMV689401 | Miconazole | Azole | CYP51 inhibitor | Antifungals (humans)[T] | 0.23 | 0.13 (<0.03–0.25) |
| MMV1634492 | Eberconazole | Azole | CYP51 inhibitor | Antifungals (humans)[T] | 0.72 | 0.50 (0.06–4) |
| MMV637528 | Itraconazole | Azole | CYP51 inhibitor | Antifungals (humans)[T,O, S] | 1.13 | 0.05 (<0.02–0.25) \* |
| MMV344625 | Carbendazim | Benzimidazole Carbamates | Binds to β-tubulin | Antifungals (agrochemical)[N] | 1.32 | 0.5 (<0.03–2) |
| MMV637413 | Fludarabine | N/A | Purine analogue | Antiviral[O, S] | 1.63 | 16 (≧16) |
| MMV396785 | Alexidine | Biguanide | Phospholipase inhibitor | Antimicrobial[N] | 2.56 | 2 (1–4) |
| MMV003143 | Fenbendazole | Benzimidazole Carbamates | Binds to β-tubulin | Antifungals[N] | 2.66 | 2 (0.25–2) |
| MMV1634386 | Oteseconazole | Azole | CYP51 inhibitor | Antifungals (humans)[O] | 2.66 | 1 (0.25–16) |
| MMV1634491 | N/A | N/A | N/A | Antifungals[N] | 2.81 | 1 (1–4) |
| MMV1782387 | N/A | Benzimidazole Carbamates | Binds to β-tubulin | Antifungals[N] | 3.22 | 4 (0.03–0.5) |
| MMV019724 | CHEMBL548113 | N/A | N/A | Antiviral[N] | 4.87 | 4 (2 –>16) |
| MMV1634358 | Amorolfine | Morpholine | Delta(14)-sterol reductase and cholestenol Delta-isomerase inhibitor | Antifungals (humansl)[T,S] | 5.72 | 4 (4) |
| MMV1505642 | CHEMBL1528043 | N/A | N/A | Antibacterial[N] | 7.35 | 16 (≧16) |
| MMV000725 | CHEMBL1197863 | N/A | N/A | Antibacterial[N] | 8.21 | 16 (≧16) |
| MMV000043 | Tafenoquine | 8-Aminoquinoline | Disrupts microtubules | Antimalarial[O] | 12.8 | 4 (4 –>16) |

[1], The MIC50 was based on ten different *M. mycetomatis* isolates. Individual MICs can be seen in our online database at GitHub (https://github.com/OpenSourceMycetoma). Routes of administration:

[N], Not applicable or No data

[O], Oral

[S], Systemic

[T], Topical

were azoles and one was a morpholine. From the eight azoles, MMV1634362 (ravuconazole) and MMV1782224 (luliconazole) were most potent with IC50 values of 0.01 μM and <0.02 μM respectively. To determine if these 20 compounds could also inhibit the growth of other *M. mycetomatis* isolates, they were tested on nine additional isolates with a different geographical origin and genetic background based on MmySTR typing [14]. As observed in Table 2, the lowest MIC50s were obtained with the azoles MMV1634362 (ravuconazole), MMV1782224 (luliconazole), MMV1634494 (isavuconazonium), MMV637528 (itraconazole)

and the orotomide MMV689401 (olorofim) with MIC50 values of 0.01 μM, <0.03 μM, 0.04 μM 0.05 μM and 0.06 μM respectively. Potent compounds with a MIC50 value of 4 μM and below were selected to be tested *in vivo* in a *Galleria mellonella* model of *M. mycetomatis* grains.

### *In vivo* activity of the fourteen most potent compound from pandemic response box

A total of fourteen compounds with an MIC50 value of ≤ 4 μM (n = 11) and *in vitro* activity against all five species tested (carbendazim, fenbendazole and tafenoquine) were tested *in vivo* in a *M. mycetomatis G. mellonella* larvae model. None of these compounds displayed toxicity in *G. mellonella* at a concentration of 20 μM/larvae. Out of the 14 compounds tested for *in vivo* efficacy, seven compounds significantly increased larvae survival as compared to the control group on day four. These included the benzimidazole carbamates fenbendazole (Log-Rank, p = 0.0278), carbendazim (Log-Rank, p = 0.0123) and MMV1782387 (Log-Rank, p = 0.0048); the azoles ravuconazole (Log-Rank, p = 0.0266), eberconazole (Log-Rank, p = 0.0168) and luliconazole (Log-Rank, p = 0.0003); and the orotomide olorofim (Log-Rank, p = 0.0091). Only four compounds still prolonged larvae survival on day ten. These were fenbendazole (Log-Rank, p = 0.035), MMV1782387 (Log-Rank, p = 0.008), ravuconazole (Log-Rank, p = 0.025) and olorofim (Log-Rank, p = 0.044). The highest larvae survival rate on day ten was achieved with olorofim at the survival rate of 33.3%. MMV1782387 presented a survival rate of 28.9%, ravuconazole of 26.7% and fenbendazole of 24.4% (Fig 4).

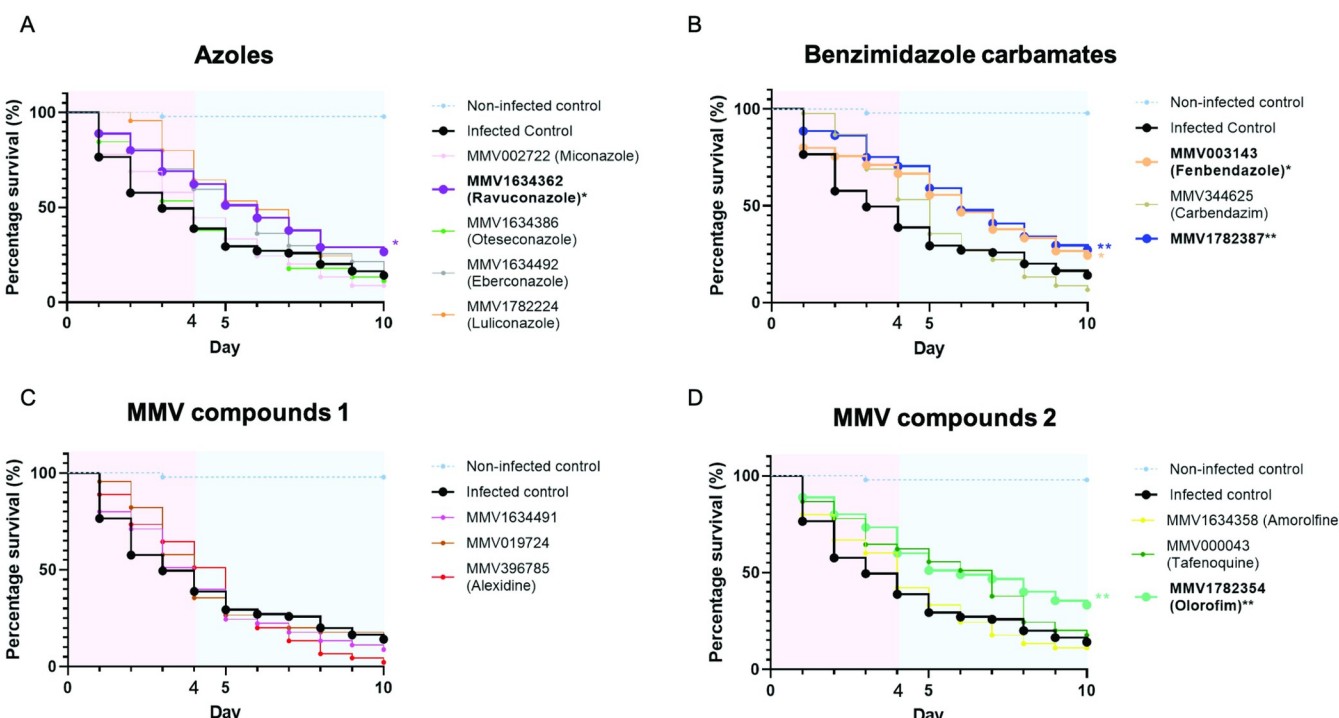

**Fig 4. Survival curves of *Galleria mellonella* larvae infected with *Madurella mycetomatis* and treated with selected compounds.** The blue dashed line in all panels represents the non-infected controls while the black line represents the infected control. Panel A displays the survival of larvae treated with azoles Miconazole, Ravuconazole, Oteseconazole, Eberconazole and Luliconazole. Panel B, the survival of larvae treated with benzimidazole carbamates Fenbendazole, Carbendazim and MMV1782387. Panel C and D displays the survival of larvae treated with the other MMV compounds MMV1634491, MMV019724, Alexidine, Amorolfine, Tafenoquine and Olorofim. Significant survival was displayed as * (0.01<p<0.05) or ** (0.001<p<0.01). Pink panels in the figure background displays the survival lines until day four of infection while the blue panel background displays day four to ten.

## Discussion

The MycetOS drug discovery program was initiated in 2018 to discover new drug candidates that could fill the drug discovery pipeline for eumycetoma treatment. In this study, we have screened 400 compounds from the MMV Pandemic Response Box in search of those active against causative agents of black-grain eumycetoma. This is the first approach taken to screen for drugs active against multiple causative agents of black-grain mycetoma. We have identified 58 compounds associated with *in vitro* activity against at least one of the fungi tested. Four compounds were able to inhibit all five causative agents. 26 inhibited all three *Madurella* sibling species, 26 inhibited *F. senegalensis* and only seven inhibited *Me. romeroi*. It did not come as a surprise that only a few compounds were active against *Me. romeroi*, as it was previously shown to be less susceptible to most antifungals tested [16]. *In vivo*, these eumycetoma causative agents form black grains. Since it is not possible to form grains *in vitro*, a host is needed for grain formation. In the past, mammalian models have been used for grain formation, but in 2015 we demonstrated that grain formation can occur in the *M. mycetomatis* infected invertebrate *G. mellonella* [22,23]. In previous studies as well in this study, we used this *G. mellonella* grain model to determine if the compounds active against hyphae *in vitro* can penetrate grains *in vivo* [10,22–24]. Four compounds, ravuconazole, olorofim, fenbendazole and MMV1782387 were able to significantly prolong larvae survival on day ten. Among these four compounds, fenbendazole was able to inhibit all five causative agents, MMV1782387 was able to inhibit all three *Madurella* sibling species and *F. senegalensis*, while ravuconazole and olorofim were only able to inhibit the growth of the three *Madurella* sibling species. One limitation of this study is that the invertebrate *G. mellonella* is genetically seen far apart from human and this can cause differences in therapeutic outcome due to difference in immune system and toxicity. *G. mellonella*, like other invertebrates, only has an innate immune response and lacks the adaptive immune response. This can cause differences in therapeutic outcome of certain drugs. For instance, when two structurally unrelated Hsp90 inhibitors were tested in a *Candida albicans* infection model in *G. mellonella*, no toxicity was noted. However, when the same drugs were tested in a mammalian infection model significant toxicity was noted and the therapeutic efficacy of the compound was lost [25]. Therefore, the next step would be to test the compounds which showed activity against the *M. mycetomatis* grain formed in *G. mellonella* in mammalian models in the future.

From our previous studies, we have determined that *M. mycetomatis* is most susceptible towards antifungals of the azole class [16,19,20,26]. It is therefore not surprising that out of the nine azoles included in the pandemic box, eight were able to inhibit *M. mycetomatis* growth below 25 μM. The same susceptibility towards azoles is also shown in *M. pseudomycetomatis* and *M. tropicana*. Out of the nine azoles, fluconazole was only able to inhibit the growth of *M. pseudomycetomatis*, not *M. mycetomatis* and *M. tropicana*. Eumycetoma causative agents belonging to the order Pleosporales in general have higher MIC50 values towards the azoles than *Madurella* species [26], which was also demonstrated in this screening. Out of the nine azoles, only eberconazole was able to inhibit *F. senegalensis* growth *in vitro*, while none of the azoles was able to inhibit *Me. romeroi* growth *in vitro* at 25 μM. It is of concern that itraconazole–the current antifungal used in eumycetoma treatment did not inhibit the growth of *F. senegalensis* and *Me. romeroi* at this concentration. The contrast in itraconazole's efficacy between the Sordariomycetes and Pleosporaleans may be due to the difference in their cell wall composition hindering access of itraconazole to its target [27]. Like other azoles, ravuconazole targets the CYP51 enzyme leading to the destabilization of the fungal cell wall. Although it shows good *in vitro* activity against several fungal species [19,28–34], clinical studies on ravuconazole have been discontinued in 2007 due to bioavailability issues [35]. Fosravuconazole, a

prodrug to ravuconazole has since been acquired by Eisai Ltd (Japan) and is currently being investigated in a randomised, double-blinded DND*i* sponsored clinical trial for mycetoma [36]. Fosravuconazole is also in clinical trials for onychomycosis [37] and Chagas disease [38]. In the clinical trial for mycetoma, the efficacy of 200 or 300 mg fosravuconazole weekly in eumycetoma patients in Sudan is compared to the current daily treatment of itraconazole at 400 mg. In this trial, only eumycetoma patients proven to be infected with *M. mycetomatis* are included while patients infected with other causative agents are excluded. Since neither *F. senegalensis* nor *Me. romeroi* growth were inhibited at 25 μM ravuconazole, further studies are needed to determine if *F. senegalensis* and *Me. romeroi* would respond to ravuconazole *in vivo*. From the five azoles tested *in vivo*, eberconazole, luliconazole and ravuconazole prolonged larvae survival on day four, while only ravuconazole was able to prolong larvae survival on day ten. Azoles have vastly different chemical properties determined by their ring structures, and that can impact their half-life, lipophilicity and subsequently their pharmacokinetic and antifungal properties [39]. Furthermore, although species dependent, most azoles are fungistatic, therefore explaining the difference between the efficacy shown on the fourth and tenth day of infection [40–42]. Further studies on eberconazole and luliconazole exploring different dosage and treatment frequency could be performed to evaluate their use in mycetoma treatment. While they may not be suitable as a sole drug to treat mycetoma, their possible use in combination treatment with other antifungals or compounds able to interfere with the grain cement material could be explored. Indeed, melanin, one of the constituents in the grain cement material was shown to lower the susceptibility against azoles *in vitro* and treating larvae with a compound inhibiting grain melanisation enhanced the therapeutic efficacy of itraconazole (Lim *et al.*, manuscript accepted) [43]. It is therefore envisioned that a similar strategy would also enhance the efficacy of other azoles. Despite the success of azoles in particular triazoles in medicine and agriculture, the common eumycetoma causative agents are still susceptible to drugs of different classes, therefore, it is also appropriate to investigate other compounds for their efficacy in treating eumycetoma.

Olorofim also demonstrated good *in vivo* efficacy in our larvae model. It resulted in 33% larvae survival on day ten. Olorofim is the leading representative of a novel class of antifungal agents called the orotomides [44]. It acts by inhibiting DHODH leading to obstruction of the pyrimidine biosynthesis pathway [21,44,45]. We have previously demonstrated that olorofim has excellent *in vitro* activity against *M. mycetomatis* [21]. Olorofim was the only DHODH inhibitor for which the activity against *M. mycetomatis* was determined. Previously, parasitic DHODH inhibitors MMV011229, MMV020591, MMV020537 and MMV020289 [10] present in the MMV Pathogen Box were screened for activity against *M. mycetomatis*. In contrast to olorofim, these DHODH inhibitors showed no activity *in vitro* against *M. mycetomatis*. This suggests that the structure-activity relationship for *M. mycetomatis* DHODH differs from that in other pathogens, thus, a certain chemical structure is needed to inhibit the *M. mycetomatis* DHODH enzyme and subsequently its growth [21,46]. Here we demonstrate that olorofim was also able to inhibit the growth in *M. pseudomycetomatis* and *M. tropicana*, however, no activity against *F. senegalensis* and *M. romeroi* was observed at a concentration of 25 μM. Next to *Madurella* species, olorofim was also demonstrated active against azole-resistant *Aspergillus* species, *Scedosporium* species and *L. prolificans* [47]. Olorofim is currently in a phase IIb clinical study to evaluate its efficacy in treatment of fungal infections in patients lacking treatment options. Patients who have resistant invasive fungal infection with limited treatment options can participate in the olorofim clinical study. This, by default, also includes eumycetoma patients. It would therefore be interesting to see if any eumycetoma patients will be included in this trial and what the treatment response will be.

**Table 3. Percentage growth of *M. mycetomatis* at 100 μM and 25 μM, IC50 and MIC50 values, *Galleria mellonella* larvae survival and *in vivo* significance of the eight benzimidazole carbamates from the Pathogen Box, Stasis Box and Pandemic Response Box tested against *Madurella mycetomatis*.**

| Benzimidazole carbamates | Growth inhibition | | IC50 (μM) | MIC50 (μM) | *In vivo* significance (p-value) | |
| --- | --- | --- | --- | --- | --- | --- |
| | 100 μM | 25 μM | | | Day 4 | Day 10 |
| MMV1782387 | 15.57 | 4.41 | 0.26 | 4 | Increase survival (0.0048) | Increase survival (0.0008) |
| MMV003143 (Fenbendazole) | 7.16 | 2.38 | 2.66 | 0.5 | Increase survival (0.0278) | Increase survival (0.035) |
| MMV344625 (Carbendazim) | 8.48 | 3.52 | 3.52 | 0.5 | Increase survival (0.0123) | No |
| MMV687730 | 7.67 | -3.36 | 11.55 | | | |
| MMV002163 (Oxibendazole) | -2.25 | 119 | 69.1 | | | |
| MMV002565 (Oxfendazole) | 76 | 87.4 | | | | |
| MMV1578842 | 93.8 | 88.1 | | | | |
| MMV003152 (Mebendazole) | 110.9 | | | | | |

The two remaining compounds that prolonged larvae survival on the tenth day were benzimidazole carbamates, namely fenbendazole and MMV1782387. While fenbendazole and MMV1782387 were able to increase larvae survival *in vivo* on day ten, the other benzimidazole carbamate carbendazim was only able to significantly increase larvae survival on day four. The *in vivo* activities shown by these three compounds has made benzimidazole carbamates a promising candidate class for the treatment of eumycetoma. Benzimidazole carbamates work by binding to β-tubulin and destabilizing microtubule formation in mammalian cells, parasites and also in fungi such as *Aspergillus nidulans* and *Fusarium graminearum* [48–50]. This impairs the motility, division and secretion process of cells resulting in cell death [50,51]. Fenbendazole is commonly used as an anthelmintic drug against gastrointestinal parasites and is currently approved for use in numerous animal species [52]. Toxicity of fenbendazole in humans is not known. Studies showing fenbendazole's anti-*Cryptococcus* [53] activity have also been reported. Carbendazim is normally used as a fungicide against a wide variety of fungal pathogens [54]. Both compounds have also shown anticancer properties [54,55]. No other information was available for compound MMV1782387. Next to these three benzimidazole carbamates, at least six other benzimidazole carbamates from the Pathogen Box, Stasis Box and the Pandemic Response box have been screened for their activity against *M. mycetomatis*. Unfortunately, most were not able to inhibit *M. mycetomatis* growth at a concentration of 25 μM (Table 3 and Fig 5) [10]. This again indicates that certain chemical properties are needed to inhibit *M. mycetomatis* growth. Stasiuk *et al.* reported an increase in the number of up-regulated genes and developmental differences when the usage of fenbendazole was compared to albendazole, mebendazole, thiabendazole, and oxfendazole in *Caenorhabditis elegans* and the ruminant parasite *Haemonchus contortus* [56]. The presence of secondary drug targets for fenbendazole was also suggested [56]. While the difference in responses between the tested benzimidazole carbamates is not yet clear in *M. mycetomatis*, it is certain that fenbendazole and MMV1782387 are promising targets to further evaluate as a treatment option both as a single treatment or in combination with other antifungals for eumycetoma.

This is the first approach taken to screen compounds for activity against five common black-grain eumycetoma causing fungi. We have identified ravuconazole, olorofim, MMV1782387 and fenbendazole as compounds able to penetrate grains and inhibit *M. mycetomatis* growth *in vivo* in the *G. mellonella* larvae model. Their activity *in vivo* has made them promising candidates to further explore in mammalian models and subsequently in mycetoma treatment and to also serve as a scaffold for medicinal chemistry optimisation in the search for novel antifungals to treat eumycetoma. The work reported in this manuscript is also a part of the Open Source Mycetoma initiative (MycetOS) created to discover new treatments for

**Fig 5. The 8 benzimidazole carbamates from the Pathogen Box, Stasis Box and Pandemic Response Box tested for *M. mycetomatis*.** Only three benzimidazole carbamates, MMV1782387, MMV0031343 (fenbendazole) and MMV344625 (carbendazim) were tested *in vivo*, while the rest wasn't able to inhibit *M. mycetomatis* growth at 25μM. Due to MMY1782387 and fenbendazole's efficacy *in vivo*, benzimidazole carbamates are a promising group of compounds to investigate as part of the MycetOS initiative in search for novel drugs to treat eumycetoma.

eumycetoma and all associated data are made available in an online database–GitHub (https://github.com/OpenSourceMycetoma). Open Source Mycetoma (MycetOS) is an open source research initiative where all data and ideas generated are openly shared, encouraging the public to participate in discussions and contribute as an equal partner as long as the principle of openwork is upheld. With the available data of more than 1300 compounds screened and the benzimidazole carbamates identified here as promising candidates, we are calling out to scientists from all disciplines to join in our discussions on GitHub and together discover novel compounds to treat eumycetoma–one of the most neglected of neglected tropical diseases.

## Supporting information

**S1 Table. All information and data of the 400 compounds tested from the Pandemic Response Box.**
(XLSX)

## Author Contributions

**Conceptualization:** Wendy van de Sande.

**Data curation:** Wilson Lim, Wendy van de Sande.

**Formal analysis:** Wilson Lim, Kimberly Eadie, Juli Smeets.

**Funding acquisition:** Annelies Verbon, Wendy van de Sande.

**Investigation:** Wilson Lim, Bertrand Nyuykonge, Kimberly Eadie, Mickey Konings, Kirandeep Samby.

**Methodology:** Wilson Lim, Kimberly Eadie, Mickey Konings, Wendy van de Sande.

**Project administration:** Wendy van de Sande.

**Resources:** Wilson Lim, Ahmed Fahal, Alexandro Bonifaz, Benjamin Perry, Kirandeep Samby, Jeremy Burrows, Wendy van de Sande.

**Software:** Wilson Lim, Benjamin Perry, Kirandeep Samby.

**Supervision:** Wilson Lim, Wendy van de Sande.

**Validation:** Wilson Lim, Kimberly Eadie, Matthew Todd.

**Visualization:** Wilson Lim, Benjamin Perry, Kirandeep Samby, Wendy van de Sande.

**Writing – original draft:** Wilson Lim.

**Writing – review & editing:** Wilson Lim, Bertrand Nyuykonge, Kimberly Eadie, Mickey Konings, Juli Smeets, Ahmed Fahal, Alexandro Bonifaz, Matthew Todd, Benjamin Perry, Kirandeep Samby, Jeremy Burrows, Annelies Verbon, Wendy van de Sande.

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
