## [Decision Letter · Decision Letter 0]

4 Nov 2021

Dear Mr. Lim,

Thank you very much for submitting your manuscript "Screening the Pandemic Response Box identified Benzimidazole carbamates, Olorofim and Ravuconazole as promising drug candidates for the treatment of eumycetoma." for consideration at PLOS Neglected Tropical Diseases. As with all papers reviewed by the journal, your manuscript was reviewed by members of the editorial board and by several independent reviewers. In light of the reviews (below this email), we would like to invite the resubmission of a significantly-revised version that takes into account the reviewers' comments. 

We cannot make any decision about publication until we have seen the revised manuscript and your response to the reviewers' comments. Your revised manuscript is also likely to be sent to reviewers for further evaluation.

Sincerely,

Husain Poonawala

Associate Editor

Kristien Verdonck

Deputy Editor

Reviewer's Responses to Questions

**Key Review Criteria Required for Acceptance?**

**Methods**

-Are the objectives of the study clearly articulated with a clear testable hypothesis stated?

-Is the study design appropriate to address the stated objectives?

-Is the population clearly described and appropriate for the hypothesis being tested?

-Is the sample size sufficient to ensure adequate power to address the hypothesis being tested?

-Were correct statistical analysis used to support conclusions?

-Are there concerns about ethical or regulatory requirements being met?

Reviewer #1: This work deals with the study of the susceptibility of several eumycetoma causative agents to a group of diverse antimicrobial substances.

-Are the objectives of the study clearly articulated with a clear testable hypothesis stated?

I consider the objectives are very clear, to determine the anti fungal activity of a series of substances, both, in vitro and in vivo

-Is the study design appropriate to address the stated objectives?

The main problem is that the authors use their own methods instead of similar standardized methods already available to test fungal isolates. I think that makes reproducibility more difficult.

-Is the population clearly described and appropriate for the hypothesis being tested?

The objectives are clearly described, but I think the methodology used is not enough to get conclusions. For instance, they use only 10 isolates of M. mycetomatis to determine the MIC50, that I consider a low amount, insufficient to get conclusions

-Is the sample size sufficient to ensure adequate power to address the hypothesis being tested?

I do not think so

-Were correct statistical analysis used to support conclusions?

NA

-Are there concerns about ethical or regulatory requirements being met?

No

Reviewer #2: 1. The manuscript under peer-review entitled, “Screening the pandemic response box identified benzimidazole carbamates, olorofim and ravuconazole as promising drug candidates for the treatment of eumycetoma.” deals with search of newer therapeutic modalities in the treatment of one of the significant neglected tropical diseases.

2. This is extensively and well-written manuscript dealing with details of newer antifungals and that too through an innovative programme i.e. Open Source Mycetoma  initiative (MycetOS).

3. The subject matter has been nicely introduced, which deals with the importance of eumycetoma vis-a-vis availability of antifungal in the tropical countries, which are usually found to be the developing countries.

4. The Open Source Mycetoma  initiative (MycetOS), a drug discovery program to discover new drug candidates was started in the year 2018 and all associated data are  made available in an online database. This fills the drug discovery pipeline for the treatment of eumycetoma. They had screened 400   compounds from the MMV Pandemic Response Box in search of those active compounds against causative agents  of black-grain eumycetoma. The drugs mentioned in the manuscript (benzimidazole carbamates, olorofim and ravuconazole) are in its continuity as a unique programme. Moreover, fenbendazole was also able to inhibit all five most frequently causative agents of eumycetoma.

Reviewer #3: Methods

-Are the objectives of the study clearly articulated with a clear testable hypothesis stated?

Yes

-Is the study design appropriate to address the stated objectives?

Yes

-Is the population clearly described and appropriate for the hypothesis being tested?

Yes

-Is the sample size sufficient to ensure adequate power to address the hypothesis being tested?

Not applicable

-Were correct statistical analysis used to support conclusions?

Yes

-Are there concerns about ethical or regulatory requirements being met?

No

**Results**

-Does the analysis presented match the analysis plan?

-Are the results clearly and completely presented?

-Are the figures (Tables, Images) of sufficient quality for clarity?

Reviewer #1: -Does the analysis presented match the analysis plan?

Not completely

-Are the results clearly and completely presented?

They are consfusing

-Are the figures (Tables, Images) of sufficient quality for clarity?

Most of them

Reviewer #2: 5. The percentage growth of causative fungi at different levels like 100 μM and 25 μM, IC50 and MIC50 values, Galleria mellonella larvae survival and in vivo significance of the antifungal from the Pathogen Box, Stasis Box and Pandemic Response Box tested, are well depicted in the text description as well as tabulated form of the manuscript.

6. The details of the Results are given minutely in detail. A total of 400 diverse-drug like compounds from the Pandemic Response box were tested in vitro to determine their potency against M. mycetomatis, M. pseudomycetomatis, M. tropicana, F. senegalensis and M. romeroi. Out of these 400 compounds screened at the concentration of 25 μM, 47, 42 and 37  compounds were able to inhibit the growth of the Sordariomycetes M. mycetomatis, M.  pseudomycetomatis and M. tropicana, respectively. The Dendrogram of the causative fungi and other depiction including the chemical formula of some of the drugs are also nicely shown.

Reviewer #3: Results

-Does the analysis presented match the analysis plan?

Yes

-Are the results clearly and completely presented?

Yes

-Are the figures (Tables, Images) of sufficient quality for clarity?

Yes, except for table 3 which is only partially visible in the PDF file.

**Conclusions**

-Are the conclusions supported by the data presented?

-Are the limitations of analysis clearly described?

-Do the authors discuss how these data can be helpful to advance our understanding of the topic under study?

-Is public health relevance addressed?

Reviewer #1: -Are the conclusions supported by the data presented?

No

-Are the limitations of analysis clearly described?

No

-Do the authors discuss how these data can be helpful to advance our understanding of the topic under study?

Yes, they do

-Is public health relevance addressed?

Yes

Reviewer #2: 7. The Discussion is very elaborative as well as exhaustive. The findings of the manuscript in the light of contemporary studies have been nicely described. This shows from the pandemic response box that newer antifungals are also equally promising.

8. The References are more than sufficient in number and doing justice with the citations vis-a-vis the finding of the manuscript. The authors have cited almost all the published papers on the issue under question i.e. the oldest to the latest ones.

Reviewer #3: Conclusions

-Are the conclusions supported by the data presented?

Yes

-Are the limitations of analysis clearly described?

Please include a limitation section in the discussion (potential variable response in humans vs larvae model, toxicity of the agents in humans, etc..).

-Do the authors discuss how these data can be helpful to advance our understanding of the topic under study?

Yes

-Is public health relevance addressed?

Yes

**Editorial and Data Presentation Modifications?**

Reviewer #1: The present work deals with the analysis of susceptibility of a series of compounds against eumycetoma agents. My comments are the following:

1. Change the title, use a more direct phrase, such as “ Studies of susceptibility of…”

2. Pgae 3. Line 81: Change “antibiotics” for “antimicrobials”, and add a reference covering this aspect

3. Page 4, line 104: Why do you used molar concentrations?

4. Page 4, line 122: Eliminate fig 1, it has nothing to do with the study

5. Page 5, line 139: Ten isolates are not enough to determine the MIC50 value. You may need at least 30

6. Page 5, line 160: Describe briefly the technique, including: age of culture, count of UFC, medium, etc.

7. Page 5, line 161: Please refer the CLSI method that you are using to determine the susceptibilities

8. Please use micrograms per ml throughout the document

Page 6, line 187: What sonifier was used? How did that affect viability? How large were the mycelial fragments?

Page 6, line 188: Four mg in what volume? Of water? It is too much inoculum, how could you inject it into the larvae without clogging the needle?

Reviewer #2: 9. Therefore keeping in view of the above-mentioned descriptions, the manuscript may be accepted for publication in our esteem journal.

Reviewer #3: (No Response)

**Summary and General Comments**

Reviewer #1: It is an interesting work, however the methods used I think are not the adequate. There are many small technical details that the authors need to clarify.

Reviewer #2: 10. This is nicely conducted study and consequently well-written manuscript with novel ideas.

Reviewer #3: The authors report on the evaluation of 400 drug-like molecules against several black grain mycetoma agents identifying several with activity against them. The effort is outstanding and relevant as it may pave the way for new mycetoma treatments in the near future. The article is very well-written and interesting. I only have a few queries and suggestions.

Please remove sentence “No new classes of antifungal agents with new modes of action are identified or evaluated.” from the author summary for being reiterative.

Please correct sentence “In Senegal, patients were given 88 terbinafine 500 mg twice daily for 24 – 48 weeks combined with surgery.” for clarity to “In a report from Senegal,..”

In table 2 Amorolfine is classified as an “Antifungals (agrochemical)”. Nonetheless, it is also employed as a topical antifungal in humans. Please modify the table to include this and specify which of the antifungals labeled as “Antifungals (humans)” use are employed systemically and which topically.

Do the authors know the reason why ravuconazole clinical studies were discontinued in 2007? If so, please state why. 

Table 3 is only partially visible in the PDF file.

PLOS authors have the option to publish the peer review history of their article (what does this mean?). If published, this will include your full peer review and any attached files.

Reviewer #1: No

Reviewer #2: No

Reviewer #3: Yes: JA Cardenas-de la Garza
---

## [Decision Letter · Decision Letter 1]

10 Jan 2022

Dear Mr. Lim,

We are pleased to inform you that your manuscript 'Screening the Pandemic Response Box identified Benzimidazole carbamates, Olorofim and Ravuconazole as promising drug candidates for the treatment of eumycetoma.' has been provisionally accepted for publication in PLOS Neglected Tropical Diseases.

Best regards,

Husain Poonawala

Associate Editor

Kristien Verdonck

Deputy Editor

Reviewer's Responses to Questions

**Key Review Criteria Required for Acceptance?**

**Methods**

-Are the objectives of the study clearly articulated with a clear testable hypothesis stated?

-Is the study design appropriate to address the stated objectives?

-Is the population clearly described and appropriate for the hypothesis being tested?

-Is the sample size sufficient to ensure adequate power to address the hypothesis being tested?

-Were correct statistical analysis used to support conclusions?

-Are there concerns about ethical or regulatory requirements being met?

Reviewer #2: In the methods whatever slight deviation was there the authors have clarified it. The arguments for deviation from the standard protocol seems to be logical and justified.

Reviewer #3: -Are the objectives of the study clearly articulated with a clear testable hypothesis stated?

Yes

-Is the study design appropriate to address the stated objectives?

Yes

-Is the population clearly described and appropriate for the hypothesis being tested?

Yes

-Is the sample size sufficient to ensure adequate power to address the hypothesis being tested?

Yes

-Were correct statistical analysis used to support conclusions?

Yes

-Are there concerns about ethical or regulatory requirements being met?

No

**Results**

-Does the analysis presented match the analysis plan?

-Are the results clearly and completely presented?

-Are the figures (Tables, Images) of sufficient quality for clarity?

Reviewer #2: The Results are now depicted as desired by the Reviewers. Whatever lacunae were there, they have been rectified. The diagrammatic representations are now very clear and no portion is obstructed in the display,

Reviewer #3: -Does the analysis presented match the analysis plan?

Yes

-Are the results clearly and completely presented?

Yes

-Are the figures (Tables, Images) of sufficient quality for clarity?

Yes

**Conclusions**

-Are the conclusions supported by the data presented?

-Are the limitations of analysis clearly described?

-Do the authors discuss how these data can be helpful to advance our understanding of the topic under study?

-Is public health relevance addressed?

Reviewer #2: The conclusion is now precise one and to the point.

Reviewer #3: -Are the conclusions supported by the data presented?

Yes

-Are the limitations of analysis clearly described?

Yes

-Do the authors discuss how these data can be helpful to advance our understanding of the topic under study?

Yes

-Is public health relevance addressed?

Yes

**Editorial and Data Presentation Modifications?**

Reviewer #2: It is properly done while revising the manuscript.

Reviewer #3: Accept

**Summary and General Comments**

Reviewer #2: Now, it is clarified and all the points have been taken care of while revising the manuscript.

Reviewer #3: Authors have addressed all the queries.

PLOS authors have the option to publish the peer review history of their article (what does this mean?). If published, this will include your full peer review and any attached files.

Reviewer #2: No

Reviewer #3: **Yes: **JA Cardenas-de la Garza

---

## [Editor Report · Acceptance letter]

20 Jan 2022

Dear Mr. Lim,

We are delighted to inform you that your manuscript, "Screening the Pandemic Response Box identified Benzimidazole carbamates, Olorofim and Ravuconazole as promising drug candidates for the treatment of eumycetoma.," has been formally accepted for publication in PLOS Neglected Tropical Diseases.

Best regards,

Shaden Kamhawi

co-Editor-in-Chief

Paul Brindley

co-Editor-in-Chief
